# Molecular Determinants of TMC Protein Biogenesis and Trafficking

**DOI:** 10.3390/ijms26136356

**Published:** 2025-07-01

**Authors:** Dedong Shao, Jinru Tan, Xiaozhi Fan, Yilai Shu, Qianhui Qu, Yi-Quan Tang

**Affiliations:** 1State Key Laboratory of Brain Function and Disorders, MOE Frontiers Center for Brain Science, Institutes of Brain Science, ENT Institute and Otorhinolaryngology Department of Eye & ENT Hospital, Fudan University, Shanghai 200032, China; 22211520031@m.fudan.edu.cn (D.S.); tanjinru@westlake.edu.cn (J.T.); 23111520011@m.fudan.edu.cn (X.F.); yilai_shu@fudan.edu.cn (Y.S.); 2NHC Key Laboratory of Hearing Medicine, Fudan University, Shanghai 200032, China; 3Shanghai Key Laboratory of Gene Editing and Cell Therapy for Rare Diseases, Fudan University, Shanghai 200031, China; qqh@fudan.edu.cn; 4Institutes of Biomedical Science, Fudan University, Shanghai 200032, China; 5Shanghai Key Laboratory of Medical Epigenetics, International Co-Laboratory of Medical Epigenetics and Metabolism (Ministry of Science and Technology), Department of Systems Biology for Medicine, Fudan University, Shanghai 200032, China

**Keywords:** TMC, deafness, AlphaFold, trafficking

## Abstract

Transmembrane channel-like (TMC) proteins are essential for hearing and balance; however, the molecular mechanisms that regulate their proper folding and membrane targeting remain poorly understood. Here, we establish *Caenorhabditis elegans* as a genetically tractable model to dissect TMC-1 trafficking by combining CRISPR knock-in strains, super-resolution microscopy, and genome-wide forward genetic screening. We show that TMC-1 robustly localizes to the plasma membrane in both neurons and muscle cells and identify a conserved valine (V803) in transmembrane domain 9 (TM9) as critical for its biogenesis and trafficking. Structural analyses guided by AlphaMissense and AlphaFold uncover two evolutionarily conserved functional hotspots, one in the extracellular loop adjacent to TM9 and the other in the TMC signature motif, which are interconnected by an evolutionarily conserved disulfide bond. Disrupting this bond in worm TMC-1 abolishes its cell-surface localization and destabilizes the mechanotransduction channel complex. Together, these findings provide a structural framework for interpreting deafness-causing mutations in human TMC1 and highlight disulfide-bond-linked hotspots as key molecular determinants of TMC protein biogenesis and trafficking.

## 1. Introduction

The senses of hearing and balance rely on hair cells in the inner ear, which detect mechanical stimuli. These specialized sensory cells convert mechanical forces, such as sound waves and head movements, into electrical signals through mechanotransduction, a process critical for auditory perception and vestibular function. The mechanotransduction machinery of hair cells comprises a macromolecular ion channel complex, including pore-forming α subunits and accessory β subunits. After decades of investigation, transmembrane channel-like protein 1 (TMC1) has been established as the pore-forming component of the transduction channel [1,2,3,4,5,6,7,8].

The TMC1 gene was first linked to hereditary hearing loss in 2002, with mutations identified in both human patients and deaf mouse models [9,10]. To date, over 90 mutations in TMC1 have been associated with various forms of hearing loss, including DFNA36 (autosomal dominant non-syndromic sensorineural deafness 36), DFNB7/11 (autosomal recessive non-syndromic sensorineural deafness 7/11), and early-onset presbycusis [11]. Consequently, TMC1 has emerged as a promising target for gene therapy and gene editing strategies aimed at treating hereditary hearing loss [12,13,14,15,16]. Nevertheless, despite its clinical significance, only a limited subset of TMC1 mutations has been functionally characterized in mouse hair cells [5,6,17,18,19,20,21,22]. Studying TMC1 is complicated by difficulties in expressing and localizing the protein to the cell surface in heterologous systems. Furthermore, experiments in mouse hair cells are time-consuming, low-throughput, and technically demanding, limiting large-scale functional studies and drug screening efforts.

To overcome these challenges, recent studies have tested various approaches, including point mutations in the transmembrane domains of TMC1/2, knockdown of endogenous genes such as UROD (a heme biosynthesis enzyme) in HEK293T cells, engineering of TMC1 with an N-terminal Fyn lipidation tag (a myristoylation/palmitoylation motif), and co-expression of TMC1 with LGR6, a GPCR chaperone naturally expressed in hair cells [23,24,25]. While these strategies have partially enabled TMC1 and TMC2 to be localized to the plasma membrane in heterologous cells and allowed for the detection of mechanosensitive currents, our own preliminary experiments failed to observe substantial membrane localization of TMC proteins, highlighting the need for further refinement and mechanistic insights into TMC1 trafficking.

The TMC gene family is evolutionarily conserved, with eight members in mammals and two in the nematode *C. elegans*. Our previous studies demonstrated that both endogenous *C. elegans* TMC-1 and exogenously expressed human TMC proteins localize to the cell surface and act as mechanosensors in worm neurons and muscle cells [26]. These findings suggest that the trafficking mechanisms directing TMC proteins to the plasma membrane in *C. elegans* closely resemble those operating in mammalian inner ear hair cells. Thus, TMC-1-expressing *C. elegans* provides a genetically tractable in vivo model to study TMC protein trafficking and to investigate the pathogenic mechanisms of deafness-associated mutations.

In mammalian inner ear hair cells, TMC1/2 proteins localize to the tips of stereocilia, as shown by advanced confocal and electron microscopy techniques [27,28,29]. However, their low abundance in stereocilia complicates efforts to determine whether these proteins are embedded in the plasma membrane or confined to nearby intracellular compartments. Although TMC1/2 and their orthologs are widely believed to function as surface-localized mechanotransduction channels, definitive evidence of their presence in the plasma membrane of any cell type or organism remains elusive. To address this gap, we used genome editing and super-resolution microscopy to examine the subcellular localization of endogenous TMC-1 in *C. elegans*, revealing robust enrichment at the plasma membrane of both neurons and muscle cells. A genome-wide forward genetic screen identified a conserved valine residue (V803) in transmembrane domain 9 (TM9) as essential for TMC-1 surface delivery. Guided by AlphaMissense and AlphaFold structural predictions, we further identified two functional hotspots connected by a conserved disulfide bond as critical determinants of TMC-1 maturation. Disruption of this disulfide linkage abolished plasma membrane localization and destabilized the TMC-1 mechanotransduction complex. Together, these results provide direct evidence supporting TMC-1 as a surface-localized channel and reveal key structural features that govern its proper folding and membrane targeting.

## 2. Results

### 2.1. MmTMC1 and MmTMC2 Do Not Localize to the Plasma Membrane of HEK 293T Cells

As the pore-forming subunits of the hair cell mechanotransduction channel, TMC1 and TMC2 proteins must localize to the cell surface. However, the mechanisms governing their cell surface trafficking remain poorly understood. Consistent with previous reports, when GFP-tagged MmTMC1 and MmTMC2 were expressed in HEK293T cells, they were predominantly retained in the cytoplasm, colocalizing with endoplasmic reticulum (ER) markers but not with plasma membrane markers (Appendix A).

Recent studies have suggested that specific point mutations in the third transmembrane domain (TM3) of MmTMC1/2, such as H353A in MmTMC1 and H406A in MmTMC2, might promote plasma membrane localization when assessed by non-permeabilized staining methods [23]. To investigate this, we generated GFP-tagged versions of these mutants and compared their localization patterns to the wild-type proteins. However, no appreciable increase in surface localization was observed for the mutants compared to the wild type (Appendix A). In contrast, fusing an 11-residue Fyn lipidation tag to the N-terminus of MmTMC1 and MmTMC2 partially promoted their plasma membrane localization in a subset of cells (Appendix A). This tag facilitates myristoylation and palmitoylation, aiding in membrane anchoring [25]. Nonetheless, the effect was variable, and a significant proportion of the proteins remained intracellular. These findings indicate that, despite the mechanically activated currents reported for these mutants, their trafficking to the plasma membrane in HEK293T cells remains inefficient. This underscores the need for further investigation into the molecular determinants of TMC1/2 trafficking.

### 2.2. Intense Plasma Membrane Localization of TMC-1 in C. elegans Neurons and Muscle Cells

To establish a tractable *in vivo* system for observing the plasma membrane localization of TMC proteins and facilitating genetic manipulation to assess how specific mutations affect their cell surface trafficking, we selected *C. elegans* as our model organism. This nematode offers a unique combination of genetic simplicity and optical transparency, enabling precise genome editing and live imaging of protein dynamics. The *C. elegans* TMC mechanotransduction channel complex consists of TMC-1/2, TMIE, and CALM-1, as revealed by cryo-electron microscopy (cryo-EM) of the native TMC-1/2 complex [8,30]. To identify cells expressing all three genes, we analyzed publicly available single-cell transcriptomic data from *C. elegans* [31]. Our analysis showed that *tmc-1, calm-1*, and *tmie (Y39A1C.1)* are expressed in OLQ and PHC sensory neurons, as well as in body wall muscle (BWM) and vulva muscle (VM) cells (Figure 1A).

Using super-resolution microscopy, we visualized *C. elegans* TMC-1 (ceTMC-1) in CRISPR knock-in worms, where GFP was tagged to the endogenous *tmc-1* locus. We found that TMC-1 was prominently localized in the plasma membrane of both muscle cells and neurons (Figure 1B–E). Remarkably, the membrane localization of TMC-1 appears to be more pronounced than that observed for a standard plasma membrane marker, underscoring the robust and specific targeting of TMC-1 to the cell membrane (Figure 1C). Furthermore, TMC-1 was detected in the plasma membrane of specialized cellular structures, including the cilia of OLQ neurons and the dendrites of PHC neurons (Figure 1D,E). These results provide clear evidence of the cell surface localization of TMC-1 in both *C. elegans* neurons and muscle cells.

### 2.3. Forward Genetic Screen in C. elegans Identifies Critical Residues for the Plasma Membrane Localization of TMC-1

To elucidate the molecular mechanisms underlying TMC-1 plasma membrane localization, we performed a forward genetic screen using a *tmc-1::GFP* knock-in strain, focusing on GFP fluorescence in VM cells. Approximately 500 animals were mutagenized, and ~30,000 F2 progeny were screened, resulting in the identification of 24 mutants with reduced or absent GFP fluorescence (Figure 2A). Whole-genome sequencing revealed that most of these mutants harbored splice-site or nonsense mutations in either *tmc-1* or *calm-1*, leading to loss of TMC-1::GFP signal (Figure 2B). Identifying *calm-1* mutations therefore corroborates our earlier finding that CALM-1 is required for both TMC-1 expression and protein stability, and mirrors recent work showing that mammalian CIB2 governs the delivery of TMC1 and TMC2 to stereocilia in cochlear hair cells [8,26].

Among the identified mutations, the missense mutation TMC-1^V803D^, which substitutes a valine with aspartate in the ninth transmembrane domain (TM9) of TMC-1, was particularly intriguing. We introduced this mutation into the parental *tmc-1::GFP* strain using CRISPR/Cas9 genome editing and confirmed the phenotype (Figure 2C). The TMC-1^V803D^ protein exhibited reduced expression and was retained in the cytoplasm across all examined cell types, including VM, OLQ, and PHC neurons, suggesting that this mutation disrupts TMC-1 biogenesis or trafficking (Figure 2D).

To determine whether the V803 residue is itself critical or whether the valine-to-aspartate substitution was simply too disruptive, we substituted V803 with several other amino acids and expressed the corresponding *tmc-1::GFP* variants in VM2 cells. While wild-type TMC-1::GFP localized robustly to the plasma membrane, all V803 mutants showed diminished expression and poor cell surface localization (Figure 2E–H). The V803S substitution caused a modest decrease in membrane localization, while V803A abolished it entirely. Substituting V803 with charged residues (aspartate or lysine) resulted in a complete loss of membrane localization and markedly reduced protein levels.

### 2.4. Human TMC1 TM9–TM10 Extracellular-Loop Mutation Hotspot and Its Conserved Equivalents in C. elegans TMC-1

To better understand why mutations at V803 impair TMC-1 trafficking, we examined the structural context of this residue. V803 is located within TM9, near the extracellular face of the membrane, adjacent to a loop region that connects TM9 and TM10. This loop includes a disordered segment flanked by two α-helices (Helix 7 and Helix 8; Figure 3A). Analysis of human deafness variant databases revealed multiple missense mutations in this extracellular loop of human TMC1. Notably, five of the mutated residues are evolutionarily conserved in *C. elegans* TMC-1 (Figure 3A,B), underscoring the potential functional relevance of this region. 

To assess how these deafness-associated mutations affect ceTMC-1 trafficking, we introduced the corresponding substitutions into ceTMC-1 and evaluated their localization when overexpressed in VM cells. Two of these mutations, P812L (Figure 3D) and C816W (Figure 3E), caused cytoplasmic retention and reduced protein levels. Mutations V828D and V829T partially impaired plasma membrane localization, whereas E834Q retained robust membrane localization (Figure 3F–H). Furthermore, co-expression of TMC-1^P812L^ or TMC-1^C816W^ with ceTMIE led to mislocalization of ceTMIE, compared to co-expression with wild-type TMC-1 (Appendix A). These results indicate that these mutations disrupt the proper trafficking of the TMC-1 mechanotransduction complex and likely contribute to hearing loss by impairing human TMC1 localization at the cell surface. 

### 2.5. AlphaMissense Reveals Two Structurally Connected Functional Hotspots in Human TMC1

Our unbiased whole-genome EMS mutagenesis screen in *C. elegans* identified a critical residue, V803, in worm TMC-1. This position aligns to V651 in human TMC1 and is strictly conserved across the TMC family. A query of hereditary deafness mutation databases revealed multiple pathogenic variants clustered around V651, suggesting that this region is particularly sensitive to alteration. To determine whether other regions of TMC1 show similar susceptibility, we applied AlphaMissense, a deep learning model built upon AlphaFold2, which predicts the pathogenic impact of every possible missense substitution in the human proteome [32]. By generating pathogenicity scores for all single amino acid variants in human TMC1, we were able to map mutational hotspots and identify domains critical for its structural integrity and function. This analysis revealed two clusters of highly pathogenic residues in human TMC1, each forming a missense mutation hotspot (Figure 4A). Within these regions, AlphaMissense assigned consistently high pathogenicity scores to specific residues—such as C515, W516, E517, and G521 (Figure 4B), as well as C664 and G665 (Figure 4C)—when mutated to most other amino acids. Intriguingly, one of these hotspots, located at the beginning of TM6 (near the extracellular loop between TM5 and TM6), is the TMC signature sequence motif (short for TMC motif), a highly conserved sequence “CWETXVGQEly(K/R)LtvXD” that defines the TMC protein family (Figure 4D). The second hotspot is located between TM9 and Helix 7 and contains residue V651, which aligns with V803 in *C. elegans* TMC-1 and is conserved across the TMC family (Figure 4E).

Strikingly, analysis of the resolved cryo-EM structure of *C. elegans* TMC-1 and TMC-2 revealed that these two pathogenic regions are structurally connected by a disulfide bond (C667–C816 in TMC-1; C594–C743 in TMC-2) (Appendix A). Using AlphaFold2 predictions [33], we extended this observation to other TMC homologs, including *Drosophila melanogaster* TMC and human TMC1–8. These predicted structures also featured disulfide bonds between equivalent cysteine residues (Appendix A–K). Notably, in the predicted structure of human TMC1 (Appendix A), C515 forms a disulfide bond with C664, both of which are identified as most likely pathogenic residues in the AlphaMissense database (Figure 4B,C). These findings underscore the evolutionary conservation and likely functional importance of this disulfide bond in TMC proteins.

### 2.6. Disruption of the Disulfide Bond Impairs TMC-1 Trafficking

To investigate the functional significance of the disulfide bond, we introduced serine substitutions at cysteine residues C667 and C816 in *C. elegans* TMC-1, disrupting the bond without introducing charge. Overexpression of these mutants (TMC-1^C667S^ and TMC-1^C816S^) in VM cells led to reduced protein levels and prominent cytoplasmic retention (Figure 5A,B), recapitulating the trafficking defects observed in GFP-tagged MmTMC proteins in HEK 293T cells (Appendix A). Furthermore, co-expression of red fluorescently labeled ceTMIE with TMC-1^C667S^ or TMC-1^C816S^ resulted in mislocalization of TMIE, indicating that disruption of the disulfide bond compromises proper assembly or localization of the TMC-1 mechanotransduction channel complex (Appendix A).

Interestingly, addition of an N-terminal Fyn tag, which directs proteins to the plasma membrane via lipid modification, was sufficient to restore membrane localization of the disulfide-bond-disrupted MmTMC1 mutants (Fyn-MmTMC1^C516S^ and Fyn-MmTMC1^C664S^) in HEK 293T cells, as shown by their colocalization with a plasma membrane marker (Appendix A). These findings suggest that the Fyn tag can override intrinsic trafficking defects caused by disulfide bond disruption, potentially through enforced membrane anchoring. However, this also highlights a caveat of the Fyn-tagged system: it may mask the impact of mutations that would otherwise impair proper protein trafficking, thereby limiting its utility in dissecting endogenous trafficking mechanisms of TMC proteins.

### 2.7. TMC-1 Disulfide Bond Formation May Involve Unknown Chaperones

Since disulfide bond formation is catalyzed by protein disulfide isomerases (PDIs) or sulfhydryl oxidases, we hypothesized that a PDI-like chaperone might facilitate disulfide bond formation and promote TMC-1 folding and trafficking. To test this, we performed feeding RNAi in *C. elegans* targeting 18 genes implicated in disulfide bond formation, including four PDI family members (*pdi-1, pdi-2, pdi-3,* and *pdi-6*), *ero-1* (which reoxidizes PDI-1), and other genes encoding sulfhydryl oxidases or PDI-like proteins (Figure 5C). Surprisingly, none of these knockdowns recapitulated the trafficking defects observed in TMC-1^C667S^ or TMC-1^C816S^ mutant strains (Figure 5D). This lack of phenotype could be attributed to insufficient knockdown efficiency of the RNAi, or it may indicate that disulfide bond formation in TMC-1 occurs independently of these known enzymes or depends on molecular chaperones that have yet to be identified.

## 3. Discussion

This study establishes *C. elegans* as a robust model for investigating TMC protein trafficking and the molecular mechanisms underlying deafness. Through an integrative approach combining structural, functional, and genetic analyses, we provide critical insights into the determinants of TMC protein folding and trafficking. Our findings reveal that two functional hotspots, interconnected by a conserved disulfide bond, are essential for the structural integrity and plasma membrane localization of TMC proteins—key prerequisites for their role in mechanotransduction.

### 3.1. The TMC Motif and Disulfide Bond: Key Determinants of TMC Protein Folding and Trafficking

The TMC signature sequence motif, a highly conserved feature among TMC family members, emerges as a key structural element in our study. This motif, together with a disulfide bond linking two distinct functional hotspots, plays a pivotal role in protein folding and trafficking. Structural analyses, integrating AlphaMissense and AlphaFold predictions with cryo-EM data, have revealed that the TMC motif and the extracellular region between TM9 and Helix 7 are connected via a disulfide bond (e.g., C515-C664 in human TMC1) that is conserved across TMC orthologs. This covalent link appears to stabilize the tertiary structure of TMC proteins, facilitating correct folding and ensuring efficient trafficking to the plasma membrane. Disruption of either the conserved motif or the disulfide linkage—through missense mutations—results in misfolding, cytoplasmic retention, and, ultimately, loss of function. These results underscore the indispensable role of the disulfide bond as a structural linchpin for TMC protein stability and trafficking. 

### 3.2. Disulfide-Bond-Connected Functional Hotspots and TMC-Associated Diseases

Mutations that affect the stability of disulfide bonds can have profound effects on protein function. In particular, changes at cysteine residues that are essential for disulfide bond formation, such as C667 and C816 in *C. elegans* TMC-1 (which correspond to C515 and C664 in human TMC1), can significantly reduce protein expression and its ability to localize to the plasma membrane. According to AlphaMissense predictions, this disulfide bond bridges two regions that are functional for pathogenic mutations. Many missense mutations near this disulfide bond in TMC1, including p.P514L, p.C515R, p.C515F, p.P660L, p.C664W, and p.S668R, have been identified and are associated with hearing loss [34,35,36,37,38]. Our research correlates these structural changes with known pathogenic mutations related to TMC1, highlighting the critical role of the C515-C664 disulfide bond in TMC1 function. Therefore, mutations that weaken the disulfide bond or its neighboring structural elements can impact the trafficking of the protein, potentially disrupting the mechanotransduction machinery in hair cells. This discovery establishes a molecular link between structural defects and the clinical presentation of deafness, enhancing our understanding of the underlying causes of hearing loss.

Building on these findings, we have also investigated the ClinVar database for missense mutations in human TMC1-8. Among these, we identified a TMC6 mutant (C540R) and a TMC8 mutant (C418W), both of which are likely to disrupt potential disulfide bonds. These mutations are associated with epidermodysplasia verruciformis, a dermatosis linked to TMC6/8 [39], and are currently classified as “Uncertain significance” due to the absence of functional validation. Our findings may provide valuable pathogenic evidence for these specific mutations and others that disrupt disulfide bonds in TMC family members, enhancing our comprehension of the molecular mechanisms underlying TMC-associated diseases.

### 3.3. ER Redox Regulation of TMC Protein Folding

The ER lumen maintains an oxidizing environment that is crucial for the formation of de novo disulfide bonds in nascent proteins. This process is facilitated by protein disulfide isomerases (PDIs), which are re-oxidized by ER oxidases such as Ero1α/β, peroxiredoxin IV, and vitamin K epoxide reductase (VKOR) [40,41,42,43]. These enzymatic pathways sustain a high GSSG:GSH ratio, which promotes the formation of native disulfide bonds and allows PDIs to resolve mispaired cysteines, ensuring that proteins acquire the correct tertiary structures [43].

Many multi-pass transmembrane proteins rely on disulfide bonds between conserved cysteines within luminal or extracellular loops to stabilize their folding and function. Disulfide bond formation often occurs co-translationally as luminal loops emerge into the ER, significantly reducing the entropy of the folding intermediates [44]. For instance, class A GPCRs require a conserved disulfide between TM3 and the second extracellular loop to exit the ER [45,46]. Disruption of this bond in rhodopsin leads to ER retention and retinitis pigmentosa [47]. Similarly, the inositol 1,4,5-trisphosphate receptor (IP3R) contains four luminal cysteine residues that form two disulfide bonds. One disulfide bond is essential for the formation of functional IP3R tetramers, while the other regulates channel activity. Oxidation of the regulatory disulfide bond by ERp46 activates IP3R, whereas its reduction by ERdj5 leads to channel inactivation [48,49].

ER quality control closely couples disulfide bond formation to folding and degradation pathways [43,44]. Chaperones such as BiP bind nascent loops to prevent premature or improper disulfide formation, while lectin chaperones (calnexin/calreticulin) coordinate glycan and disulfide maturation through ERp57 [50,51,52,53,54]. Reductive isomerases like ERdj5 and TMX4 further correct mispaired bonds, safeguarding proper folding and channeling irreparably misfolded proteins toward ER-associated degradation (ERAD) [55,56]. These processes are dynamically regulated by luminal calcium and the unfolded protein response to maintain ER redox homeostasis under stress.

By analogy to these systems, TMC proteins, which feature multiple luminal loops, likely depend on similar oxidative folding machinery. Our data show that disrupting a conserved disulfide bond (C667–C816) in *C. elegans* TMC-1 abolishes surface trafficking, highlighting the structural importance of this bond. However, targeting canonical PDIs and ER oxidases via RNAi did not fully replicate the trafficking defects, suggesting that additional, possibly TMC-specific, redox mechanisms or chaperones may be involved. Future studies should investigate ER redox dynamics during TMC biogenesis and identify key factors supporting proper folding, which may ultimately inform therapeutic strategies for trafficking-defective TMC mutants underlying hereditary deafness.

## 4. Materials and Methods

### 4.1. Experimental Worm Strains

The *C. elegans* strains used in this work are listed as Table 1.

### 4.2. Plasmids Construction

Plasmids for expression in *C. elegans* and cell lines were constructed using either the Multisite Gateway Three-Fragment cloning system (12537-023, Invitrogen, Waltham, MA, USA) or pEASY^®^-Basic Seamless Cloning and Assembly Kit (TransGen Biotech, Beijing, China). All genes amplified by PCR were sequenced to confirm accuracy.

### 4.3. Immunofluorescence and Super-Resolution Imaging

HEK293T cells were purchased from Cell Bank/Stem Cell Bank, Chinese Academy of Sciences (Shanghai, China). HEK 293T cells were cultured at 37 °C in a humidified incubator with 5% CO_2_ in Dulbecco’s Modified Eagle Medium (DMEM) supplemented with 10% fetal bovine serum (FBS) and penicillin–streptomycin. For imaging experiments, cells were seeded in 35 mm glass-bottom dishes (NEST) and transfected with plasmid DNA using polyethylenimine (PEI; Beyotime, Shanghai, China), following the manufacturer’s instructions. Live-cell staining was performed using Hoechst (Beyotime) for nuclear labeling and CellMask™ Plasma Membrane Stain (C10045, Invitrogen, USA) for membrane visualization, according to the manufacturers’ protocols. Forty-eight hours post-transfection, images were acquired using a super-resolution microscope (Olympus SpinFV-COMB, Tokyo, Japan) equipped with 60× or 100× oil-immersion objectives. Laser excitation settings were as follows: 405 nm for Hoechst, 488 nm for GFP, 594 nm for mCherry, and 647 nm for far-red signals. Identical imaging settings, including channel configuration, laser intensity, and exposure time, were maintained across experimental and control groups.

For worm imaging, live *C. elegans* were anesthetized with 25 mM muscimol and mounted between 3% agarose pads and microscope coverslips prior to imaging.

### 4.4. EMS Mutagenesis and Genetic Mapping

Ethyl methanesulfonate (EMS) mutagenesis was performed following established protocols using the *tmc-1::GFP* (TYQ28) *C. elegans* strain [57,58]. Synchronized L4-stage hermaphrodites were washed off NGM plates into 15 mL tubes with M9 buffer. After centrifugation at 700× *g* for 1 min, the supernatant was carefully removed, and the worms were washed four times with M9 buffer to eliminate residual bacteria. The worms were then resuspended in 2 mL of M9 buffer.

In a fume hood, a 0.1 M EMS stock solution was freshly prepared by adding 20 μL of EMS (Sigma-Aldrich, Waltham, MA, USA) to 2 mL of M9 buffer in a 15 mL tube, mixing gently until fully dissolved. An equal volume (2 mL) of the worm suspension was combined with the EMS solution to achieve a final concentration of 50 mM EMS. The mixture was sealed with Parafilm and incubated horizontally at 20 °C for 4 h with gentle rotation. Post mutagenesis, worms were washed five times with M9 buffer to remove residual EMS.

Mutants exhibiting altered fluorescence in VM cells were identified through screening of clonal F2 progeny, resulting in the isolation of 24 mutants. These mutants were backcrossed six times with the parental *tmc-1::GFP* (TYQ28) strain before whole-genome sequencing, which was performed by Novogene Biotechnologies, Inc. (Tianjin, China).

### 4.5. Generation of CRISPR/Cas9-Mediated V803D Mutant Strains

To generate the V803D point mutation, CRISPR/Cas9 genome editing was employed [59]. Lyophilized single-guide RNA (sgRNA) was resuspended in nuclease-free TE buffer to a concentration of 50 μM, and single-stranded oligodeoxynucleotide (ssODN) donors were resuspended in nuclease-free water to 100 μM. The injection mixture was prepared by sequentially combining the following components: 0.1 μL of 3M KCl (final concentration: 300 mM), 2.5 μL of tracrRNA (100 ng/μL), 1.4 μL of crRNA (56 ng/μL), 4.5 μL of asymmetric-hybrid double-stranded DNA (dsDNA) donors (200 ng/μL), 1 μL of pRF4 plasmid (50 ng/μL), and 0.5 μL of Cas9 protein (0.25 μg/μL). All components were centrifuged at maximum speed for 3 min prior to assembly. To form ribonucleoprotein (RNP) complexes, tracrRNA, crRNA, and Cas9 were pre-incubated at 37 °C for 10 min before adding DNA components. Premature addition of dsDNA donors was avoided, as it can reduce homology-directed repair (HDR) efficiency by over 60%. After RNP formation, 1.1 μL of ssODN donors (1 μg/μL) or 4 μg total of dsDNA donor cocktail were added. The final volume was adjusted to 10 μL with nuclease-free water. For optimal needle function, the mixture was centrifuged at 14,000 rpm for 2 min, and 8 μL of the clarified supernatant was transferred to a fresh tube. Injection needles were loaded within 15 min of preparation to maintain complex stability.

### 4.6. RNA Interference (RNAi) Feeding

RNAi feeding experiments were conducted using the *C. elegans* RNAi feeding library developed by Julie Ahringer’s group [60]. NGM agar plates supplemented with 100 μg/mL carbenicillin and 1 mM IPTG were prepared and stored at 4 °C for up to three weeks. Bacterial cultures were grown overnight at 37 °C in LB medium containing 50 μg/mL ampicillin. Approximately 400 μL of each culture was seeded onto the prepared NGM agar plates. All bacterial clones were verified by sequencing to ensure correct alignment of RNAi plasmids with their corresponding gene sequences.

After two days of bacterial growth, day 1 adult worms were bleached onto the RNAi plates. The phenotypes of the F1 progeny were assessed to determine the effects of gene knockdown. A control RNAi plate known to produce a distinct phenotype was included in each batch of experiments to confirm effective IPTG induction.

### 4.7. Protein Structure Accession Numbers

Protein structure coordinates were obtained from the Protein Data Bank (PDB) and the AlphaFold Database under the following accession numbers:

Resolved structures: *C. elegans* TMC-1: 7USW; *C. elegans* TMC-2: 8TKP. 

Predicted structures: *Drosophila melanogaster* TMC: A0A0U1QT59; Human TMC1: Q8TDI8; Human TMC2: Q8TDI7; Human TMC3: Q7Z5M5; Human TMC4: Q7Z404; Human TMC5: Q6UXY8; Human TMC6: Q7Z403; Human TMC8: Q7Z402; Human TMC1: Q8IU68.

## Figures and Tables

**Figure 1 ijms-26-06356-f001:**
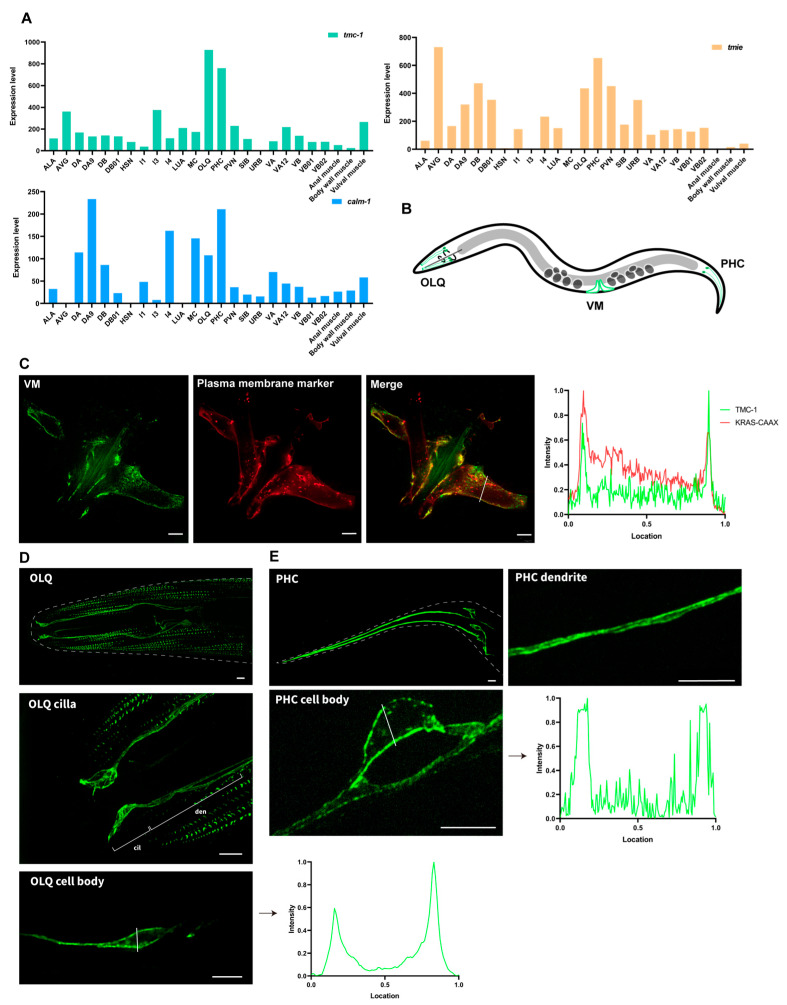
Intense plasma membrane localization of TMC-1 in *C. elegans* neurons and muscle cells. (**A**) Expression profiles of *tmc-1, calm-1,* and *tmie* (*Y39A1C.1*) in neuronal and non-neuronal cells of *C. elegans*. Data were obtained from the CeNGEN scRNA-seq dataset. (**B**) Diagram of anatomy location of OLQ, PHC, and VM in C. elegans. (**C**) Representative super-resolution fluorescence images of endogenous TMC-1 in *C. elegans* VM cells co-expressing the red fluorescently labeled plasma membrane marker, mCherry::KRAS-CAAX motif. (**D**) Representative super-resolution fluorescence images of endogenous TMC-1 in cell bodies and cilia of OLQ neurons. (**E**) Representative super-resolution fluorescence images of endogenous TMC-1 in cell bodies and dendrites of PHC neurons. Fluorescence intensity histograms for the corresponding images are shown on the right panels. Scale bars, 5 μm.

**Figure 2 ijms-26-06356-f002:**
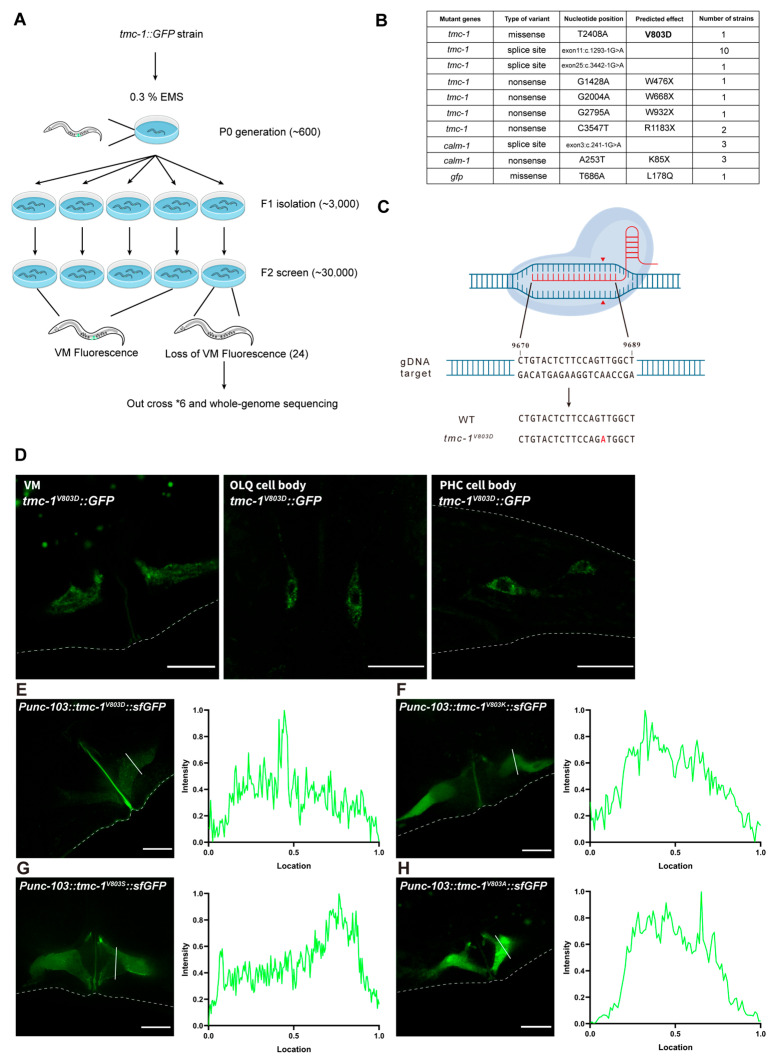
Forward genetic screen in C. elegans identifies critical residues for the plasma membrane localization of TMC-1. (**A**) Schematic diagram of the EMS mutagenesis screening for tmc-1::GFP strain. (**B**) EMS mutation mapping results showing mutant genes, types of variants, nucleotide positions, predicted effects, and the number of each mutant for 24 mutant strains losing VM fluorescence in (**A**). (**C**) Schematic diagram of CRISPR/Cas9-mediated V803D mutagenesis in the tmc-1::GFP strain. (**D**) Representative fluorescence images of endogenous ceTMC-1^V803D^ in VM cells, OLQ neurons, and PHC neurons. (**E**–**H**) Representative fluorescence images for VM cells overexpressing ceTMC-1^V803D^ (**E**), ceTMC-1^V803K^ (**F**), ceTMC-1^V803S^ (**G**), and ceTMC-1^V803A^ (**H**) mutants. Fluorescence intensity histograms for the corresponding images are shown on the right panels. Scale bars, 10 μm.

**Figure 3 ijms-26-06356-f003:**
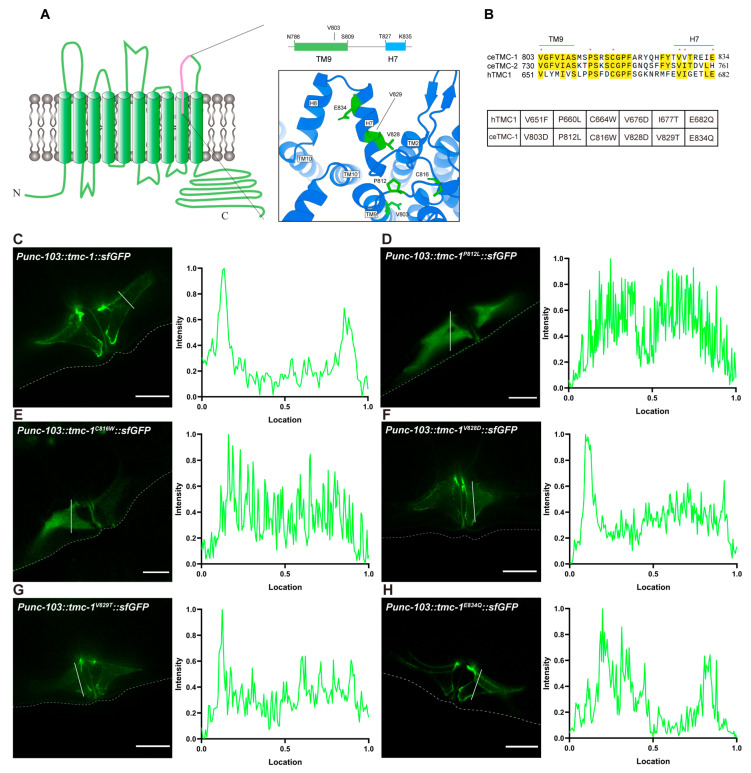
Deafness mutation hotspot in the extracellular loop between TM9 and TM10 of TMC-1. (**A**) Schematic representation of the ten-transmembrane helix topology of *C. elegans* TMC-1 (ceTMC-1) and the structure of TM9 and the adjacent extracellular loop. Residues that are conserved between ceTMC-1 and human TMC1 (hTMC1) are indicated in green. (**B**) Sequence alignment of residues 803-834 of ceTMC-1 with ceTMC-2 and hTMC1. Conserved residues identified by MUSCLE alignment are highlighted in yellow. Selected conserved residues are indicated by asterisks and listed below. (**C**–**H**) Representative fluorescence images for VM cells overexpressing WT ceTMC-1 (**C**), ceTMC-1^P812L^ (**D**), ceTMC-1^C816W^ (**E**), ceTMC-1^V828D^ (**F**), ceTMC-1^V829T^ (**G**), and ceTMC-1^E834Q^ (**H**) mutants. Fluorescence intensity histograms for the corresponding images are shown on the right panels. Scale bars, 10 μm.

**Figure 4 ijms-26-06356-f004:**
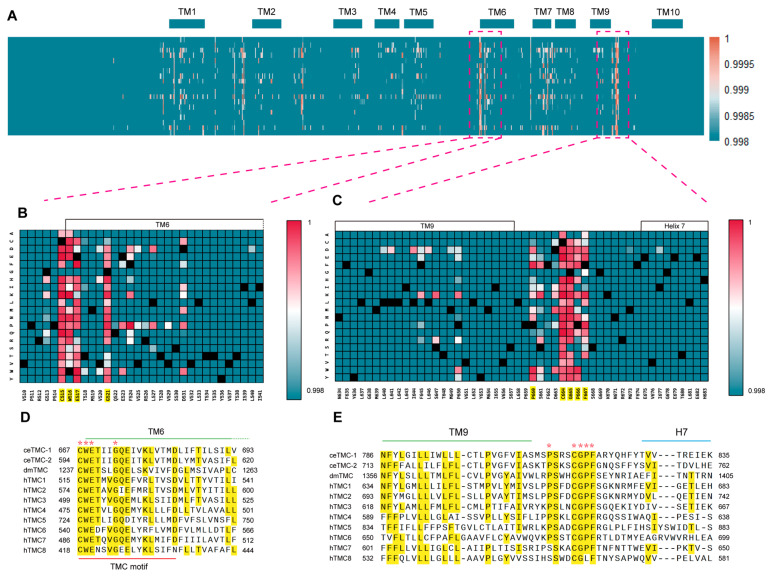
AlphaMissense predictions of two functional hotspots in human TMC1. (**A**) Heatmap of AlphaMissense scores for hTMC1. This heatmap displays the AlphaMissense pathogenicity scores for all possible amino acid substitutions at each position in the full-length hTMC1 protein sequence. The Y-axis represents the 19 alternative amino acid substitutions (excluding the wild-type), while the X-axis represents the full-length protein sequence. The color scale indicates the AlphaMissense-predicted pathogenicity scores, with darker colors representing more pathogenic substitutions (1: most pathogenic). (**B**,**C**) Enlarged heatmaps of the two hotspot regions with the highest predicted pathogenicity in hTMC1. Panel **B** focuses on residues 510–541, while panel **C** focuses on residues 634–683. The color scale matches that of panel **A**, with black bars denoting the wild-type residues. (**D**,**E**) Sequence alignment of conserved functional hotspots across species, including *C. elegans* TMC-1/2, *Drosophila* TMC, and human TMC1-8. Yellow highlights in the alignment indicate conserved residues, as determined by MUSCLE alignment. Asterisks in **D**,**E** indicate the hotspot residues, which are highlighted in yellow in **B**,**C**.

**Figure 5 ijms-26-06356-f005:**
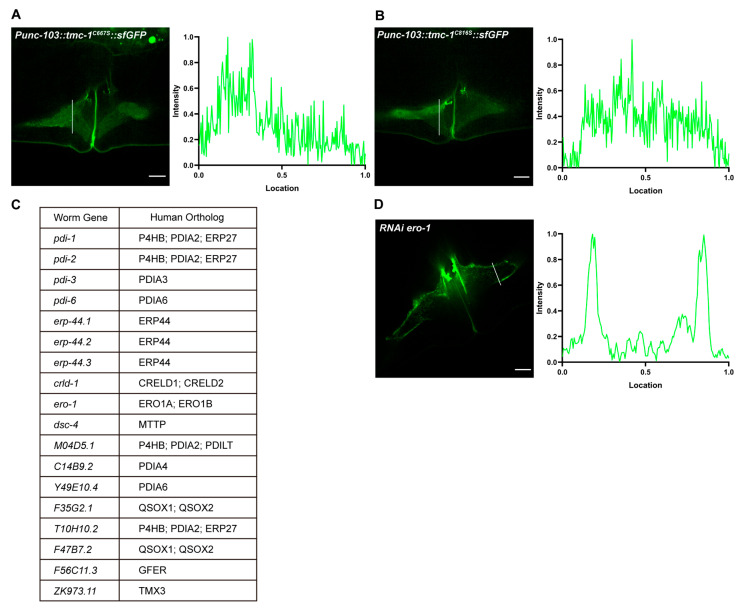
Mutations disrupting the disulfide bond impair TMC-1 trafficking. (**A**,**B**) Representative fluorescence images for VM cells overexpressing ceTMC-1^C667S^ (**A**) and ceTMC-1^C816S^ (**B**) mutants. Fluorescence intensity histograms for the corresponding images are shown on the right panels. (**C**) A list of all 18 genes screened in RNAi experiments. (**D**) Representative fluorescence images of VM cells in the *tmc-1::GFP* strain following RNAi feeding. The example shown is the knockdown of ero-1. Scale bars, 5 μm.

**Table 1 ijms-26-06356-t001:** *C. elegans* strains used in this work.

Experimental Worm Strains	Source	Identifier
*tmc-1(ok1859)*	CGC	TYQ15
*yqt1* *[tmc-1::GFP;Prps-0::HygR::unc-54 3’UTR]*	This study	TYQ28
*yqt5[tmc-1::GFP;Prps-0::HygR::unc-54 3’UTR]; tmc-1^V803D^*	This study	TYQ159
*yqt6[tmc-1::GFP;Prps-0::HygR::unc-54 3’UTR]; tmc-1(splicing exon11:c.1293-1G>A)*	This study	TYQ160
*yqt10[tmc-1::GFP;Prps-0::HygR::unc-54 3’UTR]; tmc-1^R1183*^*	This study	TYQ164
*yqt12[* *t* *mc-1::GFP;Prps-0::HygR::unc-54 3’UTR]; tmc-1^W668*^*	This study	TYQ166
*yqt13[tmc-1::GFP;Prps-0::HygR::unc-54 3’UTR]; calm-1(splicing exon3:c.241-1G>A)*	This study	TYQ167
*yqt16[tmc-1::GFP;Prps-0::HygR::unc-54 3’UTR]; calm-1^K85*^*	This study	TYQ170
*yqt21[tmc-1::GFP;Prps-0::HygR::unc-54 3’UTR]; GFP^A110V^*	This study	TYQ175
*yqt24[tmc-1::GFP;Prps-0::HygR::unc-54 3’UTR]; tmc-1^W476*^*	This study	TYQ178
*yqt25[tmc-1::GFP;Prps-0::HygR::unc-54 3’UTR]; tmc-1(splicing exon25:c.3442-1G>A)*	This study	TYQ179
*yqt29[tmc-1::GFP;Prps-0::HygR::unc-54 3’UTR]; tmc-1^W932*^*	This study	TYQ183
*yqtEx138[Punc-103e::cetmc-1^V803S^::* *sfGFP* *(50ng/μ); Pmyo-2::mCherry(5ng/μl); pcDNA3.1(to total 150ng/μl)]*	This study	TYQ277
*yqtEx139[Punc-103e::cetmc-1^V803K^::* *sfGFP* *(50ng/μl); Pmyo-2::mCherry(5ng/μl); pcDNA3.1(to total 150ng/μl)]*	This study	TYQ278
*yqtEx148[Punc-103e::cetmc-1^P812^L::* *sfGFP* *(50ng/μl); Pmyo-2::mCherry(5ng/μl); pcDNA3.1(to total 150ng/μl)]*	This study	TYQ289
*yqtEx157[Punc-103e::cetmc-1^C816W^::* *sfGFP* *(50ng/μl); Pmyo-2::mCherry(5ng/μl); pcDNA3.1(to total 150ng/μl)]*	This study	TYQ365
*yqtEx158[Punc-103e::cetmc-1^E834Q^::* *sfGFP* *(50ng/μl); Pmyo-2::mCherry(5ng/μl); pcDNA3.1(to total 150ng/μl)]*	This study	TYQ366
*yqtEx166[Punc-103e::cetmc-1^V803D^::* *sfGFP* *(50ng/μl); Pmyo-2::mCherry(5ng/μl); pcDNA3.1(to total 150ng/μl)]*	This study	TYQ374
*yqtEx175[Punc-103e::cetmc-1^V829T^::* *sfGFP* *(50ng/μl); Pmyo-2::mCherry(5ng/μl); pcDNA3.1(to total 150ng/μl)]*	This study	TYQ384
*yqtEx177[Punc-103e::cetmc-1^P812L^::* *sfGFP* *(50ng/μl);Ptmc-2-2kb::ctmie::m* *C* *herry(50ng/μl); Pmyo-2::mCherry(5ng/μl); pcDNA3.1(to total 150ng/μl)]*	This study	TYQ386
*yqtEx178[Punc-103e::cetmc-1^C816W^::* *sfGFP* *(50ng/μl);Ptmc-2-2kb::ctmie::mCherry(50ng/μl); Pmyo-2::mCherry(5ng/μl); pcDNA3.1(to total 150ng/μl)]*	This study	TYQ387
*yqtEx179[Punc-103e::cetmc-1^C816S^::* *sfGFP* *(50ng/μl); Pmyo-2::mCherry(5ng/μl); pcDNA3.1(to total 150ng/μl)]*	This study	TYQ408
*yqtEx180[Punc-103e::cetmc-1^C667R^::* *sfGFP* *(50ng/μl); Pmyo-2::mCherry(5ng/μl); pcDNA3.1(to total 150ng/μl)]*	This study	TYQ409
*yqtEx182[Ptmc-2-2kb::ctmie::mCherry(50ng/μl); Pmyo-2::GFP(2.5ng/μl); pcDNA3.1(to total 150ng/μl)); tmc-1(ok1859)*	This study	TYQ411
*yqtEx183[Punc-103e::cetmc-1^C816S^::* *sfGFP* *(50ng/μl);Ptmc-2-2kb::ctmie::mCherry(50ng/μl);Pmyo-2::GFP(2.5ng/μl); pcDNA3.1(to total 150ng/μl)]; tmc-1(ok1859)*	This study	TYQ412
*yqtEx184[Punc-103e::cetmc-1^V828D^::* *sfGFP* *(50ng/μl);Ptmc-2-2kb::ctmie::mCherry(50ng/μl); Pmyo-2::mCherry(5ng/μl); pcDNA3.1(to total 150ng/μl)]*	This study	TYQ413
*yqtEx187* *[* *Punc-103e::m* *C* *herry* *::KRAS-CAAX* *(40ng/μl); Pmyo-2::GFP(2.5ng/μl); pcDNA3.1(to total 150ng/μl); yqt1[tmc-1::GFP;Prps-0::HygR::unc-54 3’UTR]* *]*	This study	TYQ416

## Data Availability

The data presented in this study are available upon request from the corresponding author.

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
