# Peer review of "Molecular Determinants of TMC Protein Biogenesis and Trafficking"

_ijms, 2025, doi:10.3390/ijms26136356_

Round 1
Reviewer 1 Report
Comments and Suggestions for Authors
The manuscript is an incomplete early draft not suitable for review let alone publication. For example, there are no references, figures are not included in the main text, the introduction text is duplicated in the introduction, and Table A1 contains placeholders rather than real data.
Author Response
Comments 1: The manuscript is an incomplete early draft not suitable for review let alone publication. For example, there are no references, figures are not included in the main text, the introduction text is duplicated in the introduction, and Table A1 contains placeholders rather than real data.
Response 1: We sincerely apologize for the oversight in our initial submission. The issues you’ve identified, such as the absence of references and figures within the main text, were inadvertently introduced during the process of adapting our manuscript to the journal’s template. We have since corrected these errors and ensured that the complete version of the manuscript now includes all necessary references and figures integrated into the main text.
Regarding the mention of Table A1 containing placeholders, we regret any confusion this may have caused. This was an oversight during the template adaptation process, and we have removed any unnecessary placeholders, including Table A1, from the final version of the manuscript. Our study did not require the inclusion of such a table, and we have taken steps to ensure that only relevant data and tables are presented in the final submission.
We appreciate the reviewer’s feedback and are committed to addressing all concerns to ensure the manuscript meets the highest standards for publication.
Reviewer 2 Report
Comments and Suggestions for Authors
Transmembrane channel-like protein 1 (TMC1) is an evolutionarily conserved protein associated with human hearing loss, and has been a target for gene therapy to treat hereditary hearing loss. However, there were technical difficulties to properly express the protein in heterologous systems. In this manuscript Shao et al. characterized C. elegans TMC-1 protein and established C. elegans TMC-1 as a model to study human TMC1. They first showed that TMC-1 is localized plasma membrane in diverse tissues and cells in C. elegans. By EMS mutagenesis screen and prediction using AlphaMissense they further identified specific motifs and residues that are essential for the membrane localization of TMC-1. This study provides insights into the localization of the TMC1 protein in a molecular level.
Minor comments
1. There are duplicated paragraphs in the Introduction section.
2. This manuscript has some issues with C. elegans nomenclature. "C. elegans" and gene names should be italicized. Additionally, in Materials and Methods, allele names and genotypes need to be double-checked. Especially "Cas9Is" is misleading. For CRISPR strains people should use lab code, in this case "yqt". "Is" should be used for integrated transgenes.
3. Figure 1A is difficult to interpret. It was hard to compare the expression levels of the genes in individual cells/tissues. It might be helpful if the three genes are separated into independent bars.
4. In line 171, the gene "tmie" is not a C. elegans gene. Also, it says the genes are highly expressed in body wall muscle (BWM), but the expression is very low in Figure 1A. The authors need to clarify this.
5. In Figure 1B-D, the membrane localization of TMC-1::GFP is not clear in the images. It might be helpful if there are histograms, which are shown in other figures. Also, bright field images could help readers unfamiliar with C. elegans anatomy understand the data. Moreover, membrane localization of GFP can vary across focal planes. Colocalization with a membrane reporter with red fluorescence might help.
6. In line 192, it is not surprising that a lot of nonsense mutations in the tmc-1 gene decreased the expression and localization of TMC-1::GFP, because the mutations prevent GFP to be expressed. They also identified mutations in calm-1, but did not describe about the mutation. If calm-1 mutations affect the expression and localization of TMC-1, it is possible that cam-1 is upstream of tmc-1 or CALM-1 forms a complex with TMC-1. This might be worth mentioning in the results or discussion section.
7. In line 194, the authors tried to rescue the low expression of tmc-1::GFP caused by mutations in the tmc-1 gene by expressing tmc-1::GFP transgene. This experiment is fundamentally flowed for several reasons. First, they are basically overexpressing the same reporter and showing that GFP is restored. Second, because they used the tmc-1::GFP CRISPR knock-in strain, they cannot rescue tmc-1 nonsense mutants with wild-type tmc-1. Third, to validate the causal mutations, they should have created identical mutations in wild type by CRISPR or generated back mutations in the mutants by CRISPR. This section should be removed from the manuscript.
8. In line 240, "(Figure 4F-H)" should be "(Figure 3F-H)".
9. In Figure 3C and E, the dashed lines indicating the cross-sections are not visible.
10. In line 334, it is hard to make strong conclusions from negative results obtained from RNAi, because the effects of RNAi are variable or very low sometimes. Their conclusion might be misleading.
11. In line 461, the CRISPR method, the original paper that optimized CRISPR protocol should be cited in the text.
Author Response
Response to Reviewer #2:  
Transmembrane channel-like protein 1 (TMC1) is an evolutionarily conserved protein associated with human hearing loss, and has been a target for gene therapy to treat hereditary hearing loss. However, there were technical difficulties to properly express the protein in heterologous systems. In this manuscript Shao et al. characterized C. elegans TMC-1 protein and established C. elegans TMC-1 as a model to study human TMC1. They first showed that TMC-1 is localized plasma membrane in diverse tissues and cells in C. elegans. By EMS mutagenesis screen and prediction using AlphaMissense they further identified specific motifs and residues that are essential for the membrane localization of TMC-1. This study provides insights into the localization of the TMC1 protein in a molecular level.
Many thanks for your detailed and constructive review. We are grateful for your comments. During the revision, we have addressed all your comments, concerns and questions and we are confident that, with your helpful review, we have significantly improved the manuscript.
Comments 1: There are duplicated paragraphs in the Introduction section.
Response 1: We apologize for the duplication in the Introduction section. This was indeed a mistake that occurred during the process of pasting our original manuscript into the journal’s template. We have carefully reviewed the manuscript and deleted the duplicated paragraphs. We have ensured that the Introduction now presents a cohesive and non-repetitive narrative.
Comments 2: This manuscript has some issues with C. elegans nomenclature. "C. elegans" and gene names should be italicized. Additionally, in Materials and Methods, allele names and genotypes need to be double-checked. Especially "Cas9Is" is misleading. For CRISPR strains people should use lab code, in this case "yqt". "Is" should be used for integrated transgenes.
Response 2: We apologize for the inconsistencies in the use of C. elegans nomenclature. The mistakes regarding character style occurred during the process of pasting our original manuscript into the journal’s template, where most of the italicized words were inadvertently converted to regular text. We have thoroughly revised the entire manuscript to ensure that “C. elegans” and all gene names are correctly italicized.
For the allele names and genotypes mentioned in the Materials and Methods section, we have followed your valuable advice and made the necessary revisions. We have updated the naming of CRISPR strains to use the lab code “yqt” instead of the potentially misleading “Cas9Is”.
We have made these revisions in the highlighted version of the manuscript (see Materials 4.1). We appreciate your guidance in helping us adhere to the correct nomenclature and maintain the accuracy and clarity of our work.
Comments 3: Figure 1A is difficult to interpret. It was hard to compare the expression levels of the genes in individual cells/tissues. It might be helpful if the three genes are separated into independent bars.
Response 3: We have reorganized Figure 1 to separate the three genes into independent bars. This new presentation should make it easier for readers to compare the expression levels across different conditions.
Comments 4: In line 171, the gene "tmie" is not a C. elegans gene. Also, it says the genes are highly expressed in body wall muscle (BWM), but the expression is very low in Figure 1A. The authors need to clarify this.
Response 4: We appreciate your careful review and acknowledge the points raised regarding the gene “tmie” and its expression levels. Upon reevaluation, we realize that “tmie” is not the correct gene name for C. elegans. The correct gene name is “Y39A1C.1”, and its mammalian counterparts are referred to as “tmie”. To avoid confusion and maintain accuracy, we have added the original gene name “Y39A1C.1” to the manuscript (see line 132 in the highlighted version), and we have also noted that “tmie” is used in previous literature for C. elegans [1].
Additionally, we have removed the term “highly” from the sentence describing the expression levels of tmc-1, calm-1, and tmie in OLQ and PHC sensory neurons, as it may have been misleading given the data presented in Figure 1A (see line 132 in the highlighted version). We have revised the sentence to more accurately reflect the expression levels observed.
We thank you for your insightful comments and for helping us to improve the accuracy and clarity of our manuscript.
Comments 5: In Figure 1B-D, the membrane localization of TMC-1::GFP is not clear in the images. It might be helpful if there are histograms, which are shown in other figures. Also, bright field images could help readers unfamiliar with C. elegans anatomy understand the data. Moreover, membrane localization of GFP can vary across focal planes. Colocalization with a membrane reporter with red fluorescence might help.
Response 5: Thank you for your valuable feedback on Figure 1B-D. In response to your suggestions, we have taken the following actions:
- Histograms: We have incorporated histograms into Figure 1 to provide a quantitative representation of the membrane localization of TMC-1::GFP, as you recommended.
- Schematic Diagram: To assist readers unfamiliar with C. elegans anatomy, we have added a schematic diagram that delineates the anatomical locations of OLQ neurons, PHC neurons, and VM cells. This should help provide a better understanding of the spatial context of our data.
- Colocalization with a Membrane Reporter: We have added new data to Figure 1 to confirm the membrane localization of endogenous TMC-1::GFP in vulva muscles. We have expressed a plasma membrane marker—mCherry::KRAS-CAAX motif—in the vulva muscle cells. This colocalization analysis helps to validate the membrane localization of TMC-1::GFP across different focal planes. Notably, the membrane localization of TMC-1 appears to be even more pronounced than that of the plasma membrane marker, indicating a robust and specific targeting of TMC-1 to the cell membrane (see lines 135-141 in the highlighted version).
We believe these revisions will significantly improve the figure and make the data more accessible to all readers.
Comments 6: In line 192, it is not surprising that a lot of nonsense mutations in the tmc-1 gene decreased the expression and localization of TMC-1::GFP, because the mutations prevent GFP to be expressed. They also identified mutations in calm-1, but did not describe about the mutation. If calm-1 mutations affect the expression and localization of TMC-1, it is possible that cam-1 is upstream of tmc-1 or CALM-1 forms a complex with TMC-1. This might be worth mentioning in the results or discussion section.
Response 6: Thank you for your insightful observation regarding the effects of nonsense mutations in the tmc-1 gene and the potential implications for CALM-1. We agree that the relationship between calm-1 mutations and the expression and localization of TMC-1::GFP is an important aspect to consider. Our earlier work has provided biochemical evidence to demonstrate that CALM-1 forms a complex with TMC-1 [2]. Additionally, the direct interaction between CALM-1 and TMC-1 is corroborated by the recent cryo-EM structure of the C. elegans TMC-1/TMIE/CALM-1 complex [3]. To address your suggestion, we have added the following sentence to our results section (see lines 164-167 in the highlighted version): “Identifying calm-1 mutations therefore corroborates our earlier finding that CALM-1 is required for both TMC-1 expression and protein stability, and mirrors recent work showing that mammalian CIB2 governs the delivery of TMC1 and TMC2 to stereocilia in cochlear hair cells[8, 26]”. We believe this additional information strengthens the discussion and clarifies the significance of the calm-1 mutations in relation to tmc-1.
Comments 7: In line 194, the authors tried to rescue the low expression of tmc-1::GFP caused by mutations in the tmc-1 gene by expressing tmc-1::GFP transgene. This experiment is fundamentally flowed for several reasons. First, they are basically overexpressing the same reporter and showing that GFP is restored. Second, because they used the tmc-1::GFP CRISPR knock-in strain, they cannot rescue tmc-1 nonsense mutants with wild-type tmc-1. Third, to validate the causal mutations, they should have created identical mutations in wild type by CRISPR or generated back mutations in the mutants by CRISPR. This section should be removed from the manuscript.
Response 7: We have removed the flawed rescue experiment section from the manuscript as you suggested. Thank you for your feedback.
Comments 8: In line 240, "(Figure 4F-H)" should be "(Figure 3F-H)".
Response 8: Thank you for pointing out the error. We have corrected the reference from “(Figure 4F-H)” to “(Figure 3F-H)” in line 210 of the highlighted version.
Comments 9: In Figure 3C and E, the dashed lines indicating the cross-sections are not visible.
Response 9: Thank you for your feedback. We have addressed the issue with the visibility of the dashed lines in Figure 3C and E by replacing the figures with higher resolution versions. The problem has been resolved.
Comments 10: In line 334, it is hard to make strong conclusions from negative results obtained from RNAi, because the effects of RNAi are variable or very low sometimes. Their conclusion might be misleading.
Response 10: Thank you for your insightful comment. We have revised the conclusion to reflect a more cautious interpretation of the RNAi results, acknowledging the variability and potential limitations of RNAi effects. The revised statement can be found at lines 329-332 in the highlighted version. We appreciate your guidance in ensuring our conclusions are accurate and not misleading.
Comments 11: In line 461, the CRISPR method, the original paper that optimized CRISPR protocol should be cited in the text.
Response 11: Thank you for your advice. We used the optimized CRISPR protocol modified from Craig Mello’s protocol [4]. The references have been updated accordingly.
References
- Jiang, Q.; Zou, W.; Li, S.; Qiu, X.; Zhu, L.; Kang, L.; Muller, U., Sequence variations and accessory proteins adapt TMC functions to distinct sensory modalities. Neuron 2024, 112, (17), 2922-2937 e8.
- Tang, Y. Q.; Lee, S. A.; Rahman, M.; Vanapalli, S. A.; Lu, H.; Schafer, W. R., Ankyrin Is An Intracellular Tether for TMC Mechanotransduction Channels. Neuron 2020, 107, (1), 112-125 e10.
- Jeong, H.; Clark, S.; Goehring, A.; Dehghani-Ghahnaviyeh, S.; Rasouli, A.; Tajkhorshid, E.; Gouaux, E., Structures of the TMC-1 complex illuminate mechanosensory transduction. Nature 2022, 610, (7933), 796-803.
- Dokshin, G. A.; Ghanta, K. S.; Piscopo, K. M.; Mello, C. C., Robust Genome Editing with Short Single-Stranded and Long, Partially Single-Stranded DNA Donors in Caenorhabditis elegans. Genetics 2018, 210, (3), 781-787.
Reviewer 3 Report
Comments and Suggestions for Authors
- A significant portion of the introduction appears to be duplicated and should be revised for clarity and conciseness.
- There are misnumbered figures in the main text—what is labeled as Fig. 4f–h should correctly be Fig. 3f–h.
- The hotspot residues discussed in Fig. 4 are not clearly highlighted. Clear visual representation of these residues is essential for the reader’s understanding.
- The inclusion of predictions involving conserved disulfide bonds (Figure 5) appears unnecessary, as their presence and role are already well-established in the literature. This figure does not seem to add new insight and could be reconsidered.
- The Results and Discussion sections lack a clear logical flow. The manuscript shifts abruptly from membrane localization of TMC-1 to AlphaMissense predictions, and then to conserved cysteine residues. Reorganizing the sections to follow a more coherent narrative would improve readability.
- The manuscript would benefit from more thorough proofreading, as several issues in formatting, figure numbering, and writing persist throughout.
- Higher-resolution images with clearer labeling would significantly enhance the presentation.
- I could not access the supporting information, so I am unable to comment on the data or findings presented there.
Author Response
Response to Reviewer #3:
Many thanks for your detailed and constructive review. We are grateful for your comments. During the revision, we have addressed all your comments, concerns and questions and we are confident that, with your helpful review, we have significantly improved the manuscript.
Comments 1: A significant portion of the introduction appears to be duplicated and should be revised for clarity and conciseness.
Response 1: We apologize for the duplication in the Introduction section. This was indeed a mistake that occurred during the process of pasting our original manuscript into the journal’s template. We have carefully reviewed the manuscript and deleted the duplicated paragraphs. We have ensured that the Introduction now presents a cohesive and non-repetitive narrative.
Comments 2: There are misnumbered figures in the main text—what is labeled as Fig. 4f–h should correctly be Fig. 3f–h.
Response 2: Thank you for pointing out the error. We have corrected the reference from “(Figure 4F-H)” to “(Figure 3F-H)” in line 210 of the highlighted version.
Comments 3: The hotspot residues discussed in Fig. 4 are not clearly highlighted. Clear visual representation of these residues is essential for the reader’s understanding.
Response 3: Thank you for your valuable feedback regarding the clarity of the hotspot residues in Fig. 4. We agree that clear visual representation is essential for the reader’s understanding. In response to your suggestion, we have highlighted the hotspot residues in yellow in panels B and C, and added asterisks in panels D and E to indicate these residues. We believe these changes enhance the clarity and facilitate a better understanding of the data.
Comments 4: The inclusion of predictions involving conserved disulfide bonds (Figure 5) appears unnecessary, as their presence and role are already well-established in the literature. This figure does not seem to add new insight and could be reconsidered.
Response 4: Thank you for your feedback on Figure 5. We appreciate your insight that the predictions involving conserved disulfide bonds might not add substantial new insights, given their well-established presence and role in the literature. In response to your comments, we have decided to move Figure 5 to the supplementary material, allowing us to maintain detailed information for interested readers while streamlining the main text. We believe this visual summary provides readers with an intuitive framework to understand our research results, helping them to better grasp the content of the entire manuscript.
Additionally, we would like to provide some context that supports our decision to include this figure, even as supplementary material. Although the two conserved cysteines, one in the CWET sequence in the S5–S6 loop and one in the S9–S10 loop, are close enough to form a disulfide bond and show coevolution in analysis of over 3000 TMC sequences, suggesting close apposition, their precise role remains unclear. Previous studies have shown that hair cells expressing wild-type TMC1 constructs showed no change in the amplitude of sensory transduction currents following cysteine modification reagent MTSET application, suggesting that native cysteines in the WT sequence were either unable to react with MTSET or inconsequential for whole-cell currents. Therefore, in our study, we have explored the role of this conserved disulfide bond in TMC protein biogenesis and trafficking, which we believe adds value to our investigation.
Comments 5: The Results and Discussion sections lack a clear logical flow. The manuscript shifts abruptly from membrane localization of TMC-1 to AlphaMissense predictions, and then to conserved cysteine residues. Reorganizing the sections to follow a more coherent narrative would improve readability.
Response 5: We thank the reviewer for highlighting the abrupt transitions in our Results and Discussion. In the revised manuscript, we have added a brief transitional paragraph (lines 233-242) to guide the reader from our membrane localization studies into the AlphaMissense analysis. This new text reads:
“Our unbiased whole genome EMS mutagenesis screen in C. elegans identified a critical residue, V803, in worm TMC‑1. This position aligns to V651 in human TMC1 and is strictly conserved across the TMC family. A query of hereditary deafness mutation databases revealed multiple pathogenic variants clustered around V651, suggesting that this region is particularly sensitive to alteration. To determine whether other regions of TMC1 show similar susceptibility, we applied AlphaMissense, a deep learning model built upon AlphaFold2, which predicts the pathogenic impact of every possible missense substitution in the human proteome [32]. By generating pathogenicity scores for all single amino acid variants in human TMC1, we were able to map mutational hotspots and identify domains critical for its structural integrity and function.”
By inserting this paragraph, we establish a clear, stepwise narrative:
- EMS screen in C. elegans → identification of V803
- Evolutionary conservation → correspondence to human V651
- Database survey → clustering of deafness mutations around V651
- AlphaMissense analysis → unbiased mapping of pathogenicity across the entire protein
This ordering ensures that each experiment logically motivates the next, improving readability and coherence.
Comments 6: The manuscript would benefit from more thorough proofreading, as several issues in formatting, figure numbering, and writing persist throughout.
Response 6: Thank you for your feedback. We have addressed the issues that arose from pasting the original manuscript, including the incorrect character styles. We have revised all gene and species names to be italicized and fixed other issues as suggested by the reviewers.
Comments 7: Higher-resolution images with clearer labeling would significantly enhance the presentation.
Response 7: Thank you for your suggestion regarding the resolution and labeling of the figures. We acknowledge the issue with the figures’ resolution, which was caused by incorrect settings in Word. We have corrected this by replacing the figures with higher-resolution versions. To further enhance the presentation, we have also improved the visual accessibility of the figures by using larger fonts and clearer forms. We believe these changes address your concerns and significantly improve the overall quality of the figures.
Comments 8: I could not access the supporting information, so I am unable to comment on the data or findings presented there.
Response 8: Thank you for bringing this to our attention. We understand the importance of providing access to supporting information for a comprehensive review. We have followed the journal’s guidelines and the practices of other published papers in MDPI, which typically do not include supplementary figures in the initial review version. However, in response to your request, we have now included the supplementary figures at the end of the results section in the highlighted version for your review. We hope this addition facilitates a more thorough evaluation of our data and findings..